# The Evolving Landscape of Therapeutics for Epilepsy in Tuberous Sclerosis Complex

**DOI:** 10.3390/biomedicines11123241

**Published:** 2023-12-07

**Authors:** Giovanni Vitale, Gaetano Terrone, Samuel Vitale, Francesca Vitulli, Salvatore Aiello, Carmela Bravaccio, Simone Pisano, Ilaria Bove, Francesca Rizzo, Panduranga Seetahal-Maraj, Thomas Wiese

**Affiliations:** 1Neuroscience and Rare Diseases, Discovery and Translational Area, Roche Pharma Research and Early Development (pRED), F. Hoffmann–La Roche, 4070 Basel, Switzerland; 2Department of Translational Medical Sciences, Child and Adolescent Neuropsychiatry, University of Naples Federico II, 80138 Naples, Italy; gaetano.terrone@unina.it (G.T.); carmela.bravaccio@unina.it (C.B.);; 3School of Medicine and Surgery, University of Naples Federico II, 80138 Naples, Italy; sam.vitale@studenti.unina.it; 4Department of Neurosciences and Reproductive and Dental Sciences, Division of Neurosurgery, University of Naples Federico II, 80138 Naples, Italyilariabove90@gmail.com (I.B.); 5Department of Neuroscience “Rita Levi Montalcini”, University of Turin, 10126 Turin, Italy; rizzofrancesca839@gmail.com; 6Department of Neurosurgery, King’s College Hospital NHS Foundation Trust, London SE5 9RS, UK

**Keywords:** tuberous sclerosis complex, epilepsy in TSC, TSC treatment options, epilepsy medications, seizure control, epilepsy management, mTOR inhibitors, neuropsychiatric symptoms in TSC, anti-seizure medications, everolimus, cannabidiol, TANDs

## Abstract

Tuberous sclerosis complex (TSC) is a rare multisystem genetic disorder characterized by benign tumor growth in multiple organs, including the brain, kidneys, heart, eyes, lungs, and skin. Pathogenesis stems from mutations in either the TSC1 or TSC2 gene, which encode the proteins hamartin and tuberin, respectively. These proteins form a complex that inhibits the mTOR pathway, a critical regulator of cell growth and proliferation. Disruption of the tuberin–hamartin complex leads to overactivation of mTOR signaling and uncontrolled cell growth, resulting in hamartoma formation. Neurological manifestations are common in TSC, with epilepsy developing in up to 90% of patients. Seizures tend to be refractory to medical treatment with anti-seizure medications. Infantile spasms and focal seizures are the predominant seizure types, often arising in early childhood. Drug-resistant epilepsy contributes significantly to morbidity and mortality. This review provides a comprehensive overview of the current state of knowledge regarding the pathogenesis, clinical manifestations, and treatment approaches for epilepsy and other neurological features of TSC. While narrative reviews on TSC exist, this review uniquely synthesizes key advancements across the areas of TSC neuropathology, conventional and emerging pharmacological therapies, and targeted treatments. The review is narrative in nature, without any date restrictions, and summarizes the most relevant literature on the neurological aspects and management of TSC. By consolidating the current understanding of TSC neurobiology and evidence-based treatment strategies, this review provides an invaluable reference that highlights progress made while also emphasizing areas requiring further research to optimize care and outcomes for TSC patients.

## 1. Introduction

Tuberous sclerosis complex (TSC) is a rare genetic disorder that is characterized by the growth of typically benign tumors in multiple different organs throughout the body, including the brain, kidneys, heart, eyes, lungs, and skin. It is caused by mutations in either the *TSC1* gene on chromosome 9q34.13 or the *TSC2* gene on chromosome 16p13.3, which code for the proteins hamartin and tuberin, respectively [1]. Hamartin and tuberin proteins bind together to form a complex that functions as a critical tumor growth suppressor. When either the *TSC1* or *TSC2* gene is mutated in a way that prevents normal production of hamartin or tuberin, the growth-suppressing function of the complex is disrupted, leading to overactivation of mTOR pathway signaling and uncontrolled cell growth, which results in tumor formation [2].

The incidence or rate of occurrence of TSC is estimated to be around 1 in 6000 births, with approximately two-thirds of all cases arising spontaneously from new genetic mutations and one-third being inherited in an autosomal dominant pattern from a parent [3].

The clinical signs and symptoms of TSC can be highly variable but most commonly involve benign tumor growths as well as other abnormalities affecting the brain, the skin, the kidneys, the heart, and the lungs (Figure 1).

Epilepsy is a very frequent manifestation, developing in up to 90% of TSC patients, and it is often the first presenting symptom that brings parents to seek medical evaluation of a child [4]. In the brain, subependymal nodules, cortical tubers, and subependymal giant cell astrocytomas (SEGAs) are among the most common lesions that can be detected through medical imaging like CT scans or MRI scans [5]. Furthermore, TSC is often associated with neuropsychiatric disorders, collectively referred to as TSC-associated neuropsychiatric disorders (TANDs). TANDs encompass a range of behavioral, psychiatric, intellectual, academic, neuropsychological, and psychosocial symptoms and may include neurodevelopmental disorders such as autism spectrum disorder and intellectual disability.

In the kidneys, renal manifestations include angiomyolipomas—which are benign tumors containing blood vessels, smooth muscle cells, and fat cells—kidney cysts, and risk of renal cell carcinoma.

Cardiac rhabdomyomas are benign tumors that are seen in up to 60% of TSC cases, mostly in newborns or in pediatric age [6].

Pulmonary lymphangioleiomyomatosis is a progressive disease affecting the lungs; it occurs almost exclusively in females and causes cystic destruction of lung tissue [7].

Most common skin manifestations include facial angiofibromas, hypomelanotic macules, and shagreen patches. Other clinical presentations occurring in TSC include retinal hamartomas, gingival fibromas, which are benign growths on the gums, and ungual fibromas [5].

TSC is currently diagnosed through a combination of clinical evaluation using established diagnostic criteria [3]. Characteristic CT, MRI, or ultrasound findings as well as genetic testing for mutations in the *TSC1* or *TSC2* gene are decisive for establishing the diagnosis [8].

Approximately 10 to 15 percent of individuals with TSC have no identified causal mutation even after undergoing comprehensive conventional molecular diagnostic evaluation [9]. A negative result on molecular genetic testing does not preclude a diagnosis of TSC, as up to 15% of patients meeting definite diagnostic criteria for TSC have no detectable mutations in the *TSC1* or *TSC2* gene [9]. 

Treatment focuses on managing symptoms, especially controlling seizures, treating skin lesions, and preserving kidney function. The medications everolimus and sirolimus, which are inhibitors of the mechanistic target of rapamycin (mTOR) pathway, have shown promise in reducing the size of brain, kidney and lung tumors associated with TSC [10].

## 2. The Tuberin–Hamartin Complex: A Critical Regulator of mTORC1 and p53 Signaling

As previously mentioned, the tuberin–hamartin heterodimer functions as a key negative regulator of mTORC1 signaling and cell growth. The tuberin–hamartin complex acts as a key upstream negative regulator of the mechanistic target of rapamycin complex 1 (mTORC1), which controls many cellular growth processes. In addition, the complex positively regulates the p53 tumor suppressor pathway. mTORC1 is a growth regulatory kinase complex that integrates inputs from growth factors, nutrients, energy status, and stress [11]. When active, mTORC1 phosphorylates substrates S6K and 4E-BP1 to promote protein synthesis, ribosome biogenesis, nutrient transport, and autophagy inhibition [12]. A key upstream activator of mTORC1 is the small GTPase Rheb, which binds to and stimulates mTORC1 kinase activity [13]. Tuberin acts as a GTPase activating protein (GAP) towards Rheb, increasing Rheb’s intrinsic GTP hydrolysis activity to convert Rheb into its inactive GDP-bound state [14]. By inactivating Rheb, tuberin effectively inhibits mTORC1 signaling. Hamartin is required for tuberin’s GAP activity, with loss of either protein leading to mTORC1 hyperactivation [15]. In addition, the tuberin–hamartin complex regulates the trafficking and localization of mTORC1 pathway components to lysosomal surfaces, where Rheb activation of mTORC1 takes place [16,17]. Through these mechanisms, the tuberin–hamartin complex is a critical negative regulator of mTORC1 and cell growth. In addition to suppressing mTORC1 signaling, the tuberin–hamartin complex helps maintain the p53 tumor suppressor in an active state. As a transcription factor, p53 is activated in response to various cell stresses and induces expression of genes involved in cell cycle arrest, apoptosis, senescence, and DNA repair [18]. p53 activity is negatively regulated by the E3 ubiquitin ligase Mdm2, which promotes p53 proteasomal degradation [19]. Multiple lines of evidence demonstrate that tuberin binds directly to p53 and protects it from Mdm2-mediated ubiquitination and degradation by inhibiting Mdm2 access to key p53 ubiquitination sites [20,21,22,23]. By preserving p53 protein levels, tuberin allows enhanced p53 transcriptional program activation [24,25,26]. Consistently, loss of tuberin expression in TSC models reduces p53 protein levels and target gene expression [27,28,29,30]. These findings establish tuberin as a critical positive regulator of p53 function. Biallelic inactivation of either TSC1 or TSC2 dysregulates both mTORC1 and p53 signaling cascades, permitting uncontrolled cell growth and hamartoma development [2,31]. mTORC1 hyperactivation increases anabolic processes and inhibits catabolic pathways like autophagy [32]. Concurrent p53 destabilization reduces cell cycle arrest, apoptosis, and senescence [33]. The synergistic overactivation of cell growth pathways and suppression of tumor suppressive responses promotes the benign proliferative lesions characteristic of TSC [34,35] (Figure 2).

Anti-seizure and anti-epileptogenic drugs for Tuberous Sclerosis Complex: Established, Novel, and Emerging Therapies.

## 3. Overview of Conventional ASMs Used in TSC

Epilepsy affects around 85–90% of patients with TSC [4], and seizures tend to be refractory to treatment with anti-seizure medications (ASMs) [36]. Infantile spasms and focal seizures are the most common seizure types, with onset typically in the first few years of life [4]. The refractory nature of TSC-associated seizures contributes significantly to morbidity and mortality in this patient population [37].

Therefore, there is a critical need for novel ASMs that control seizures (anti-ictogenic effect) but also for anti-epileptogenic interventions that prevent the onset of the disease and also improve the related pathological outcomes after the disease is diagnosed (disease-modifying treatments) [38]. In this sense, early detection of biomarkers for epilepsy, developmental delay, and autism symptoms in infants with TSC is crucial in order to design novel anti-epileptogenic treatments [39].

The current evidence on the efficacy and safety of ASMs commonly used for the treatment of epilepsy in patients with TSC is summarized below (Table 1).

*Vigabatrin* is recommended as a first-line treatment for infantile spasms in TSC patients by European guidelines [40]. One small RCT in 22 infants found VGB led to complete seizure freedom in all 11 treated patients compared to only 5 of 11 given hydrocortisone [41]. Based on these results, a randomized placebo-controlled trial called PREVeNT was designed to further investigate vigabatrin as a preventative anti-epileptogenic drug in infants with TSC. The PREVeNT trial (NCT02849457) enrolled 110 TSC infants up to 5 months of age with no history of clinical seizures or epileptiform activity on EEG. Participants were randomly assigned 1:1 to receive vigabatrin or placebo. After 12 months of treatment or until initial clinical seizure occurrence, the blinded vigabatrin/placebo phase was completed. Participants then entered a long-term, open-label follow-up phase for continued collection of efficacy and safety data while receiving vigabatrin treatment per clinical guidelines. The primary outcome measure was time to onset of clinical seizures from enrollment through 12 months of treatment. Secondary endpoints included time to first EEG epileptiform activity, number of seizure-free days, and neurodevelopmental measurements. By comparing vigabatrin initiated prior to seizure onset versus placebo, the PREVeNT trial aimed to provide evidence regarding the preventative efficacy, safety, and impact on developmental trajectories of preemptive anti-epileptogenic therapy in TSC infants. Studies report seizure freedom rates of 27–89% for infantile spasms and 25–46% for focal seizures, with ≥50% responder rates of 73–88% [42,43,44,45]. However, 30–40% of patients develop visual field defects, which may be irreversible [46]. Tolerance due to the GABAergic mechanism is also a concern [47]. Still, recent evidence suggests high-dose VGB may reduce relapse risk [48]. Overall, VGB remains an important initial treatment option for TSC-associated seizures.

*Valproate (VPA)* enhances GABAergic inhibition and modulates sodium and calcium channels [49]. A study in 60 TSC patients found that 70% had a ≥50% responder rate with VPA [36]. However, VPA carries risks of hepatotoxicity and pancreatitis, among other potential side effects [49].

*Lamotrigine (LTG)* inhibits voltage-gated sodium channels [49]. A retrospective study in 57 TSC patients reported that 79% had a ≥50% responder rate with LTG, including 42% who were seizure-free [50]. Gradual titration is required to minimize the risk of serious skin reaction, including Stevens–Johnson syndrome (SJS) and toxic epidermal necrolysis (TEN) [49].

*Levetiracetam (LEV)* binds synaptic vesicle protein SV2A, a synaptic vesicle protein that transports neurotransmitters into vesicles and interacts with synaptotagmin to mediate calcium-regulated exocytosis [49]. A retrospective study of 20 TSC patients found that 40% had a ≥50% responder rate with LEV [51]. Psycho-behavioral adverse events are a concern, especially in those with cognitive impairment [49].

*Oxcarbazepine (OXC)* blocks voltage-gated sodium channels [49]. A retrospective study reported that 67% had a ≥50% responder rate in 16 TSC patients on OXC [36].

*Carbamazepine (CBZ)* stabilizes inactive sodium channel conformations [49]. A retrospective study found that 67% had a ≥50% responder rate with CBZ in 29 TSC patients [36]. Aplastic anemia is a rare but serious risk.

*Clobazam (CLB)* activates GABA-A receptors [49]. A retrospective study in 29 TSC patients reported that 69% had a ≥50% responder rate [52]. Sedation, physical dependence, and tolerance may occur.

*Lacosamide (LCS)* is thought to reduce the spread of seizure activity through voltage-gated sodium (NaV) channels. LCS selectively enhances slow inactivation of NaV channels, preventing excessive firing of action potentials. A retrospective study by Geffrey et al. (2015) evaluated the efficacy of lacosamide (LCS) as adjunctive therapy for treating refractory focal epilepsy in 46 patients with TSC. LCS was started at low doses and titrated up over time. About half of the patients (48%) were responders, experiencing at least a 50% reduction in seizure frequency. The responder rate was comparable to other anti-seizure medications used for TSC. Side effects were relatively mild. The authors concluded that LCS appears effective and well-tolerated for refractory focal seizures in TSC [53].

### 3.1. Perampanel

Perampanel (PER) is a selective AMPA receptor antagonist approved as an add-on ASM for treating focal seizures from 4 years of age. In a retrospective study, 38 children (mean age 4 years) treated with PER at a pediatric neurology clinic were followed. Seizure frequency and adverse events were assessed at 6 and 12 months. Responders were defined as having ≥50% seizure reduction. The responder rates were 44% at 6 months and 31% at 12 months. Complete seizure freedom occurred in 13% and 10%, respectively. Interestingly, patients with tuberous sclerosis complex (TSC) or Dravet syndrome had high response rates (67%). In refractory spasms, the 6-month response rate was 40% but dropped to 13% by 12 months. Retention rates were 61% at 6 months and 52% at 12 months [54].

### 3.2. Everolimus

Everolimus was discovered on Rapa Nui (Easter Island) during a search for new antifungal agents [55]. It was isolated from the actinobacterium Streptomyces hygroscopicus and found to have potent antifungal and immunosuppressive properties. Everolimus is a derivative of sirolimus (rapamycin), another natural product isolated from bacteria on Easter Island. Compared to sirolimus, everolimus has a shorter half-life and improved bioavailability, making it more suitable for clinical use. Everolimus selectively inhibits mTORC1 but not mTORC2. It forms a high affinity complex with FK506 binding protein 12 (FKBP12), which binds to and inhibits the kinase activity of mTORC1. This prevents downstream mTORC1 signaling, leading to inhibition of T lymphocyte activation and proliferation. The anti-epileptic properties of everolimus arise from prolonged opening of calcium and potassium channels in neurons, mediated by increased expression of voltage-gated potassium channel Kv1.1 in cortical and hippocampal neurons and decreased expression of AMPA receptors. Everolimus also exhibits neuroprotective effects by modulating synaptic plasticity, regulating neuronal cell death pathways, and impacting neurogenesis [56].

The “EXIST” (EXamining everolimus In a Study of TSC) trials aimed to evaluate everolimus efficacy and safety in different TSC-associated manifestations. Subependymal giant cell astrocytoma (SEGA) develops in up to 20% of patients with TSC, causing obstruction [3]. SEGAs arise from subependymal glial precursor cells that abnormally proliferate due to disinhibition of the mTOR pathway [57]. These slow-growing, WHO grade I tumors are located along the lateral ventricles and can cause obstructive hydrocephalus from ventricular obstruction [58]. The historical paradigm for first-line therapy was surgical resection; however, disease recurrence and residual tumor often necessitated adjuvant treatment modalities. Contemporary guidelines recommend mTOR inhibition as the preferred initial therapeutic intervention, with the notable caveat that symptomatic lesions may warrant alternate approaches as first-line agents [59]. The EXIST-1 trial was an open-label, phase 3 study evaluating the efficacy and safety of everolimus for treatment of SEGA in 117 patients with TSC [60]. Patients aged 0–65 years with serial MRI confirmation of SEGA were randomized to receive everolimus titrated to a target through a concentration of 5–15 ng/mL (*n* = 78) or a placebo (*n* = 39). The primary endpoint was SEGA response rate, defined as ≥50% reduction in volume at 6 months on MRI compared to baseline. Secondary endpoints included change in seizure frequency and skin lesion response rate. Results showed 35% of everolimus patients had ≥50% reduction in SEGA volume versus none with the placebo. The most common adverse events were mouth ulcers, stomatitis, seizures, and fever. This provided initial evidence for mTOR inhibition as a targeted therapy for SEGA in TSC. The EXIST-2 trial was a randomized, double-blind, placebo-controlled phase 3 trial assessing everolimus for angiomyolipoma in 118 TSC patients ≥18 years with at least one angiomyolipoma ≥3 cm [61]. Patients were randomized to everolimus 10 mg daily (*n* = 79) or a placebo (*n* = 39). The primary endpoint was angiomyolipoma response rate (≥50% reduction in total volume). Key secondary endpoints were time to angiomyolipoma progression and skin lesion response. The angiomyolipoma response rate was 42% for everolimus versus 0% for placebo (*p* < 0.0001). Everolimus significantly delayed angiomyolipoma progression versus placebo (*p* < 0.0001) and had a higher skin lesion response rate (26% vs. 0%, *p* = 0.0002). Adverse events were consistent with the known safety profile of everolimus. Infections occurred in 65% of everolimus and 72% of placebo patients. Everolimus demonstrated efficacy for angiomyolipoma reduction in TSC, with a manageable safety profile. As mentioned, epilepsy affects up to 90% of patients with TSC and is often refractory to multiple anti-seizure medications [4,61]. mTOR activation may play a key role in TSC-associated epilepsy [62,63]. The EXIST-3 trial was a randomized, double-blind, placebo-controlled phase 3 study assessing the efficacy and safety of the mTOR inhibitor everolimus as adjunctive therapy for treatment-resistant focal seizures in TSC [64]. The study randomized 366 TSC patients aged 2–65 years with ≥16 seizures during an 8-week baseline period despite one to three anti-epileptic drugs. Patients were randomized to a placebo (*n* = 119), low-exposure everolimus targeting 3–7 ng/mL (*n* = 117), or high-exposure everolimus targeting 9–15 ng/mL (*n* = 130). The primary endpoint was change in seizure frequency during a 12-week maintenance period. The response rate (≥50% seizure reduction) was 15.1% for placebo, 28.2% for low-exposure everolimus (*p* = 0.0077), and 40.0% for high-exposure everolimus (*p* < 0.0001). Median seizure reduction was 14.9% for placebo, 29.3% for low-exposure everolimus (*p* = 0.0028), and 39.6% for high-exposure everolimus (*p* < 0.0001). Adverse events were consistent with the known everolimus safety profile, mainly stomatitis, diarrhea, and respiratory infections. The EXIST trials have established the efficacy and acceptable safety of everolimus for major TSC manifestations. However, questions remain regarding optimal dosing, long-term outcomes, timing of initiation, and impact on wider aspects of TSC.

### 3.3. Cannabidiol

Cannabidiol (CBD) appears to have variable effects on mTOR in different models, suggesting this interaction may be context-dependent [65,66,67]. A randomized controlled trial (RCT) by Thiele et al. demonstrated the efficacy of CBD as an adjunctive ASM in 224 patients with TSC [68]. Participants were randomized to receive a placebo or CBD at doses of 25 mg/kg/day (CBD25) or 50 mg/kg/day (CBD50). The primary outcome was the percentage change in seizure frequency over the treatment period compared to baseline. CBD25 and CBD50 groups had significantly greater reductions in TSC-associated seizure frequency relative to placebo (49% and 48% vs. 27%, *p* = 0.0009 and *p* = 0.0018, respectively). Additionally, more patients in the CBD groups achieved a ≥50% reduction in seizures (33% for CBD25, 40% for CBD50, and 15% for placebo). CBD also led to more seizure-free days and greater improvements in overall condition based on the Subject/Caregiver Global Impression of Change (S/CGIC) scale. Results were maintained over 48 weeks in an open-label extension (OLE) study. At least 6% of patients remained seizure-free during any 12-week interval.

Concerning safety and tolerability, in the RCT, adverse events (AEs) occurred in 93% of the CBD25 group and 100% of the CBD50 group compared to 95% with the placebo [69]. Common AEs included diarrhea, decreased appetite, somnolence, and hepatic enzyme elevations. Most hepatic enzyme elevations occurred with concomitant valproate use.

Based on these data, CBD has been approved as an adjunctive anti-seizure medication for seizures associated with TSC in patients from two years of age. In a recent systematic review and meta-analysis conducted by Talwar et al., the findings from key clinical trials of the FDA-approved CBD oral solution for treatment-resistant epilepsies were consolidated. Pooled analyses indicated that CBD significantly reduced seizure frequencies in pediatric patients with LGS, DS, and TSC, although CBD was associated with increased risks of adverse events, including somnolence, decreased appetite, and diarrhea [70] (Table 2).

## 4. Emerging Anti-seizure Medications

### 4.1. Basimglurant

Metabotropic glutamate receptor 5 (mGluR5) is an essential G protein-coupled receptor activated by the neurotransmitter glutamate. It is classified within the group I family of metabotropic glutamate receptors, along with mGluR1. These receptors are coupled to Gq proteins and initiate intracellular signaling cascades upon glutamate binding [69]. mGluR5 is widely expressed in the central nervous system, predominantly located on postsynaptic elements, where it exerts modulation over neuronal excitability, synaptic plasticity, and neuronal signaling [71]. Studies have highlighted roles in learning, memory, and synaptic plasticity [72]. The overactivation of mGluR5 has also been implicated in excitotoxicity, neuroinflammation, chronic pain, and epileptogenesis [73]. Furthermore, dysregulation in mGluR5 signaling has been linked to a range of neuropsychiatric disorders, including anxiety, depression, Fragile X syndrome, TSC, and autism spectrum disorder [74]. Researchers have developed both positive and negative allosteric modulators (PAMs and NAMs) that potentiate or inhibit mGluR5 activity, respectively [75,76,77]. Basimglurant represents a novel compound acting as a negative allosteric modulator of mGluR5. Basimglurant exhibits advantageous brain penetration and demonstrates high in vivo potency, further augmenting its potential as a promising therapeutic candidate [78]. A phase 2B clinical trial is underway to evaluate basimglurant as adjunctive therapy for uncontrolled seizures in pediatric, adolescent, and young adult patients with TSC. The 30-week double-blind study will determine the optimal dosing and assess the efficacy and safety of basimglurant. Patients demonstrating favorable response and tolerability will have the opportunity to continue treatment in a 52-week open-label extension. The study aims to address the need for improved therapies for uncontrolled seizures in TSC [79].

### 4.2. Ganaxolone

Ganaxolone belongs to the class of neuroactive steroids that exhibit potent modulatory effects on GABAA receptors [80]. By selectively binding to a specific site on GABAA receptors, ganaxolone enhances the receptor’s response to gamma-aminobutyric acid (GABA), the primary inhibitory neurotransmitter in the brain. Upon binding to GABAA receptors, ganaxolone increases the frequency of channel opening events, resulting in increased chloride ion influx into neurons. This hyperpolarization of neuronal membranes leads to decreased neuronal excitability.

Reddy and Rogawski demonstrated the potent anticonvulsant effects of ganaxolone in the mouse amygdala kindling model [81]. In their study, ganaxolone effectively suppressed both behavioral and electrographic seizures, underscoring its potential as a broad-spectrum anti-epileptic agent. Koenig et al. presented results from a phase 2 open-label study of adjunctive ganaxolone in 23 patients aged 2–32 years with refractory TSC-related epilepsy [82]. After 4-week titration up to 63 mg/kg/day (maximum 1800 mg/day), patients entered an 8-week maintenance period. The median percentage reduction in seizure frequency was 16.6% versus baseline; at least 50% of participants had a response rate of 30.4% or higher. The most common adverse events were somnolence, fatigue, and sedation. Post hoc analysis suggested possible superior efficacy in patients not experiencing somnolence. The authors proposed that enhancing tolerability through optimized dose titration may improve seizure control in future trials [82]. A phase 3, global, double-blind, randomized, placebo-controlled trial evaluating the efficacy and safety of adjunctive ganaxolone treatment in children and adults with epilepsy associated with TSC is currently ongoing. The study will enroll approximately 162 participants aged 1–65 years with a clinical or genetic diagnosis of TSC and refractory epilepsy, defined as failure to achieve seizure control despite an adequate trial of at least two anti-epileptic drugs.

Participants will be randomized 1:1 to receive oral ganaxolone or a matching placebo three times daily. The study consists of a 4-week prospective baseline phase, a 4-week ganaxolone/placebo titration period, and a 12-week maintenance period. The primary efficacy outcome is the percentage change in 28-day seizure frequency from baseline during the titration and maintenance periods. Secondary endpoints include 50% responder rate, clinical global impression of improvement, quality of life measures, and adverse events.

This trial aims to evaluate the efficacy and safety of adjunctive ganaxolone compared to the placebo for reducing seizure frequency in TSC patients with drug-resistant epilepsy [83]. The results could support regulatory approval of ganaxolone for TSC-associated seizures.

Non-Pharmacological Interventions for Seizure Management in Tuberous Sclerosis Complex

### 4.3. Ketogenic Diet

The ketogenic diet (KD) is a high-fat, low-carbohydrate, adequate-protein diet that has been shown to be effective and safe in the treatment of TSC-related drug-resistant epilepsy. KD can reduce seizure frequency and may improve cognition and behavior in TSC patients, playing an anti-epileptic role by inhibiting the over-activated mTOR signaling pathway and through other multi-target mechanisms involving neurotransmitters, brain energy metabolism, oxidative stress, and ion channels [84]. In a recent multicenter study on 53 children with drug-resistant epilepsy or cognitive impairment caused by TSC, KD reduced seizure rates by 51.0, 45.1, 47.1, 43.1, and 25.5% at 1, 2, 3, 6, and 12 months, respectively. In addition, 36 of the 51 patients (70.6%) with psychomotor retardation exhibited a significant improvement of cognitive function after KD therapy.

In another study, KD exhibited a good efficacy and retention rate at 3 months, with 67.7% showing a response >50% [85]. These studies confirmed that KD may be effective in the treatment of drug-resistant epilepsy associated with TSC; however, more studies on the application of KD in this rare disease group are needed.

### 4.4. Surgery

While ASMs are effective in managing seizures in many cases, some patients remain refractory to medical therapy, leading to a need for surgical interventions. Recent evidence from the literature suggests that seizure freedom was achieved in 55–60% of TSC patients who underwent different neurosurgical techniques [86].

Neurosurgical treatment of epilepsy in TSC requires careful patient selection, comprehensive preoperative evaluation, and a multidisciplinary approach. The identification of the epileptogenic zone is critical for optimizing surgical outcomes and minimizing the risk of neurological deficits. In cases where epileptogenic foci are multiple or widespread, a staged surgery approach may be necessary [87]. Predictors of favorable postsurgical outcomes include total resection of epileptic tubers, the presence of an underlying tuber, monthly seizure frequency (versus daily frequency), shorter duration of epilepsy, and age at onset of seizure after the first year of life [85,86,87,88].

Assessing the potential impact of surgery on cognitive function is of particular importance in TSC patients, as cognitive impairment is a common feature of the disorder. An individualized approach taking into account each patient’s unique clinical presentation and comorbidities is essential for achieving optimal outcomes. Neurosurgical interventions, including resective surgery, corpus callosotomy, neuromodulation techniques, and laser therapy, are valuable treatment options for managing epilepsy in TSC when medical therapy fails to provide adequate seizure control.

### 4.5. Resective Surgery

Resective surgery involves the resection of epileptogenic brain tissue to eliminate or reduce seizure activity. In patients with TSC, resective surgery is a valuable option when cortical tubers or SEGAs are identified as the primary sources of epileptogenesis. Cortical tubers, characterized by abnormal neuronal organization and connectivity, can cause both focal and generalized seizures. Resection of tubers can lead to seizure reduction or even seizure freedom in some patients. SEGAs, on the other hand, are benign tumors that may cause mass effect and seizures due to their location in the ventricles. Surgical resection of SEGAs can alleviate the mass effect and, in many cases, result in seizure control. Preoperative evaluation, including neuroimaging (brain MRI, positron emission tomography co-registered with MRI), video-electroencephalography, and neuropsychological assessment, is essential to accurately localize the epileptogenic zone (EZ) and minimize the risk of postoperative neurological deficits [40]. In complex cases, an invasive monitoring with intracranial electrodes could be mandatory to better define the EZ and the area to resect. So far, evidence from longitudinal studies has shown that resective surgery in TSC series is able to achieve a seizure freedom of 65–75% at 1 year, 57% at 5 years, and 48–51% at 10 years of follow-up [84].

### 4.6. Corpus Callosotomy

Corpus callosotomy is a procedure that involves surgical disconnection of the corpus callosum in order to prevent the spread of seizures from one hemisphere to the other, thus reducing the frequency and severity of generalized seizures. Corpus callosotomy is considered in TSC patients when focal resection is not feasible or when seizures originate from both hemispheres [89]. Several variations of callosotomy exist, including anterior, complete, and partial callosotomy, each having their own unique advantages and limitations. The choice of callosotomy technique depends on the patient’s seizure semiology, video-EEG findings, and the extent of seizure spread between the hemispheres. This technique has shown promising results especially for the treatment of drug-resistant epileptic or tonic spasms secondary to TSC [90].

### 4.7. Neuromodulation Techniques

Neuromodulation techniques offer non-destructive treatment options for patients with medication-resistant epilepsy. Two commonly utilized techniques include vagus nerve stimulation (VNS) and responsive neurostimulation (RNS).

VNS involves the implantation of a device that delivers electrical stimuli to the vagus nerve. VNS has been shown to reduce seizure frequency and improve quality of life in patients with medication-resistant epilepsy, including those with TSC. The precise mechanisms of how VNS exerts its anti-epileptic effects are not fully understood, but it is believed to modulate neuronal excitability and synchronize neural circuits associated with seizures [91]. In TSC patients, VNS should be considered as a safe and effective treatment for DRE, acting not only on seizure frequency but also improving depressive mood. However, a follow-up of at least one year should be required to predict long-term outcomes in TSC patients [92].

RNS is an innovative technique that involves the implantation of a closed-loop neurostimulation system, which continuously monitors brain activity and delivers electrical stimulation to the seizure focus. RNS uses both depth and cortical electrodes to detect and respond to abnormal epileptiform activity in real time. Once abnormal activity is detected, the device delivers electrical stimulation to interrupt seizure propagation and prevent the occurrence of overt seizures. In a small case series of five patients with TSC-related refractory epilepsy with failed surgical approaches, including VNS, resective surgery, and corpus callosotomy, RNS demonstrated promising results, reducing seizures by about 86% with a follow-up duration of 25 months [93].

### 4.8. Laser Interstitial Thermal Therapy (LITT)

Magnetic resonance–guided laser interstitial thermal therapy (MR-gLiTT) is a novel, minimally invasive treatment approach for drug-resistant focal epilepsy. A laser diode is implanted intracranially with a stereotactic robot guided approach to produce a thermal ablation [94]. MR provides a real-time monitoring of the delivery temperature, allowing the safe management of multiple lesions even in proximity to eloquent brain structures [95]. LITT is an emerging treatment modality that has shown promise in the treatment of lesions that cause drug-resistant epilepsy.

## 5. Conclusions

As of today, robust evidence from randomized controlled trials is still lacking for most therapies. While targeted treatments have advanced, a precision medicine approach tailored to each patient’s specific mutations, seizure types, and developmental stage may be needed to optimize outcomes. In summary, progress has been made in elucidating pathogenesis and expanding treatment options for the neurological manifestations of TSC, but continued research is critical to provide the strong evidence base required to develop best practice guidelines and improve quality of life for patients with this challenging disorder.

Despite advances in understanding the genetic basis and pathophysiology of TSC, there remain significant unmet needs in its management. Early diagnosis is a key challenge, as the clinical manifestations of TSC are highly variable and can be subtle, leading to delays in diagnosis and initiation of treatment. This can result in irreversible organ damage and significant morbidity. There is also a need for earlier preventive treatments to halt disease progression and improve long-term outcomes. Currently, most treatments for TSC are symptomatic and do not address the underlying disease process. Combination therapies that include medications and other therapies such as surgery, dietary therapy, or behavioral interventions could potentially offer more comprehensive management of TSC. However, the development and validation of such combination therapies require rigorous clinical trials and are currently an area of unmet need.

The current treatment options for TSC are limited, and there is a pressing need for the development of new therapies. One area of interest is the more complete inhibition of the mammalian target of rapamycin (mTOR) in the brain, as current mTOR inhibitors have limited brain penetration and may not fully inhibit mTOR activity in the brain. Selective mTOR inhibition is another promising approach that could potentially reduce the side effects associated with systemic mTOR inhibition. Further novel mechanisms of action that target other aspects of TSC pathophysiology are also needed to provide more comprehensive disease management. Genetic therapies that correct the underlying genetic defects in TSC1 or TSC2 could potentially offer a cure for TSC. Finally, therapies that selectively induce cytotoxicity in TSC1- or TSC2-null tumor cells could provide a targeted approach to eliminate TSC-associated tumors without harming normal cells. These potential therapeutic strategies represent important areas for future research in TSC treatment. 

## Figures and Tables

**Figure 1 biomedicines-11-03241-f001:**
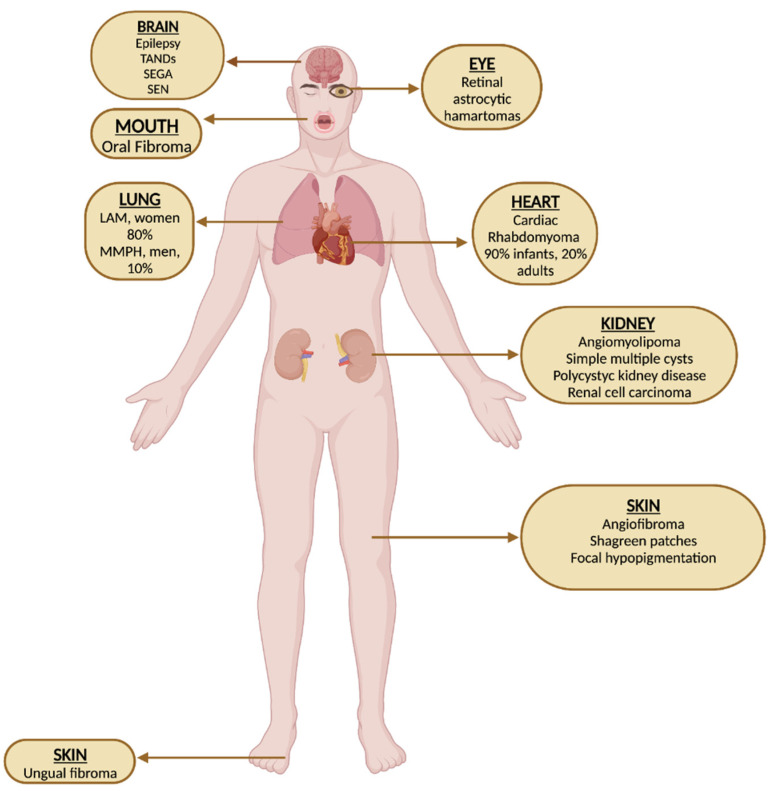
Clinical manifestations of TSC.

**Figure 2 biomedicines-11-03241-f002:**
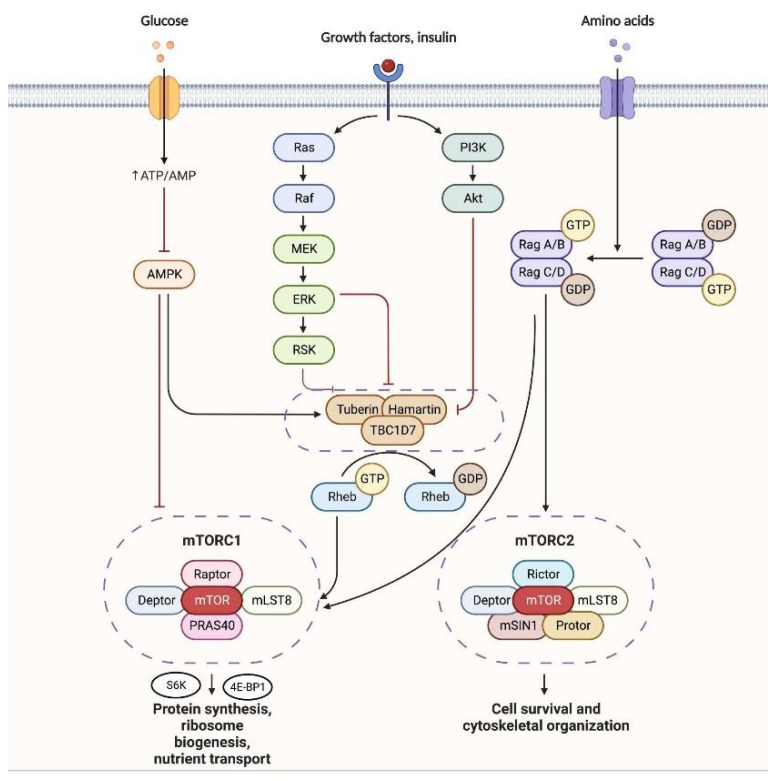
mTORC1 and mTORC2 pathway.

**Table 1 biomedicines-11-03241-t001:** Mechanism of action and adverse effects of conventional AEDs used in TSC.

Drug	Mechanism of Action	Adverse Effects
Vigabatrin	GABA-T inhibitor	Vigabatrin has been associated with retinal toxicity, resulting in permanent bilateral concentric visual field constriction. Baseline and periodic ophthalmologic examinations consisting of visual acuity and visual field testing are required to monitor for visual field defects during vigabatrin treatment. However, these evaluations may not be feasible in patients with profound intellectual disability who cannot properly participate in visual field examinations.
Valproate	Voltage-dependent sodium channel inhibitor	Treatment with valproate has been associated with risks of severe hepatotoxicity and teratogenic effects, necessitating careful consideration and monitoring in appropriate populations. Hepatotoxicity, ranging from transient asymptomatic elevations in liver enzymes to fatal hepatic failure, has been reported with valproate use. Additional adverse reactions that may occur with valproate include abdominal pain, alopecia, vision changes, amnesia, decreased appetite, weakness, impaired coordination, respiratory infections, constipation, diarrhea, and depression. Patients initiated on valproate require the routine monitoring of liver enzymes to screen for hepatotoxicity.
Lamotrigine		Lamotrigine has been associated with severe, life-threatening rash, especially in pediatric patients also receiving valproic acid. There is a risk of additional immune-mediated adverse reactions, including hepatotoxicity. Careful assessment of risks versus benefits is required when prescribing lamotrigine concurrently with valproic acid in the pediatric population. Additional adverse effects requiring monitoring include lethargy, ataxia, dysarthria, nausea, dizziness, and somnolence. A slow titration schedule over several weeks is necessary when initiating lamotrigine to minimize the risk of rash and other side effects. Patients started on lamotrigine require close follow-up to detect any emerging adverse events.
Lacosamide		Treatment with lacosamide has been associated with several adverse effects, which require consideration. Dizziness, imbalance, vomiting, double vision, nausea, vertigo, and vision changes are frequently reported side effects. As lacosamide enhances slow inactivation of voltage-gated sodium channels, caution is advised when co-administering lacosamide with other anti-epileptic drugs that act on sodium channels, as pharmacodynamic interactions may increase the risk of adverse effects. Patients initiated on lacosamide require close monitoring for the emergence of side effects, which may necessitate dosage adjustments or discontinuation.
Levetiracetam	SV2A modulator	Levetiracetam use requires careful consideration in patients with tuberous sclerosis complex given reports of worsening irritability, aggression, and other behavioral disturbances. Due to the risk of exacerbating underlying neuropsychiatric symptoms, levetiracetam is generally not recommended for TSC patients exhibiting behavioral concerns. Commonly reported adverse effects of levetiracetam include fatigue, nasal congestion, and decreased appetite. The risks of adverse neuropsychiatric reactions with levetiracetam must be weighed against the potential benefits for each individual TSC patient. Close monitoring for behavioral changes is essential if levetiracetam is prescribed to patients with a history of aggression or irritability. Overall, caution is advised when considering levetiracetam for TSC populations, given the predisposition for neuropsychiatric symptomatology.
Carbamazepine	Voltage-sensitive sodium and calcium channel inhibitor	Carbamazepine use requires careful consideration given its multiple potential adverse effects. Common side effects include dermatologic reactions, aggression, irritability, and mood lability. Carbamazepine should generally be avoided in patients with tuberous sclerosis complex exhibiting aggression, due to the risk of exacerbating behavioral symptoms. Serious dermatologic reactions such as Stevens–Johnson syndrome occur more frequently in certain genetic populations. Carbamazepine also carries black box warnings for blood dyscrasias like aplastic anemia and agranulocytosis. Additional side effects may include central nervous system depression, ataxia, diplopia, and gastrointestinal distress. As a potent cytochrome P450 enzyme inducer, carbamazepine can accelerate metabolism of other medications, resulting in decreased blood concentrations and potential loss of efficacy. The complex adverse effect and drug interaction profile warrants careful patient selection and monitoring with carbamazepine. Risks versus benefits should be weighed given the availability of alternate anticonvulsants with improved tolerability.
Oxcarbazepine		Oxcarbazepine requires careful consideration in patients with tuberous sclerosis complex and renal angiomyolipomas given its renal metabolism and elimination. Rare but serious dermatologic reactions, including Stevens–Johnson syndrome and toxic epidermal necrolysis, have occurred more frequently in patients positive for the HLA-B*1502 allele. Monitoring is warranted, as life-threatening skin conditions have been reported. Additional adverse effects requiring monitoring include electrolyte abnormalities like hyponatremia, dermatologic hypersensitivity reactions, elevations in liver enzymes, and increased susceptibility to viral infections.
Clobazam	Benzodiazepines/ GABA A receptor agonists	Clobazam requires careful consideration given the potential for paradoxical reactions, tolerance, dependence, and abuse liability. Clobazam use has been associated with increased agitation, aggression, and other behavioral adverse effects in pediatric patients, contrary to its expected pharmacological activity. The risks of paradoxical reactions and the exacerbation of underlying behavioral symptoms must be weighed against potential benefits, especially in vulnerable populations. Additional adverse effects include sedation, fever, respiratory infections, fatigue, and hypersalivation. Clobazam possesses habit-forming properties, and abrupt discontinuation may precipitate withdrawal symptoms such as anxiety, tachycardia, and tremor. The need for gradual tapering must be considered if discontinuing clobazam after prolonged use. Overall, the adverse effect profile warrants judicious use, and patients require vigilant monitoring for paradoxical reactions, signs of dependence, or withdrawal phenomena. The risks versus benefits of clobazam should be carefully evaluated on a case-by-case basis.

**Table 2 biomedicines-11-03241-t002:** Mechanism of action and adverse effects of newer AEDs used in TSC.

Drug	Mechanism of Action	Adverse Effects
Perampanel	AMPA receptor antagonists	Dizziness, somnolence, and headache are the most commonly reported adverse events with perampanel use. A higher percentage of patients reporting dizziness was observed with the increase in the perampanel dose. Aggression, blurred vision, and irritability were observed in <1% of perampanel users.
Everolimus	mTOR inhibitor	Frequently reported adverse effects include respiratory infections, stomatitis, hyperglycemia, and lipid abnormalities. Serious hematologic toxicities such as thrombocytopenia, neutropenia, and febrile neutropenia occurred in pediatric TSC patients on everolimus therapy. The risks of infection, hematologic toxicity, metabolic disturbances, and potential neurodevelopmental impairment with everolimus must be weighed carefully against the benefits of seizure reduction in TSC populations.
Cannabidiol	Cannabinoid receptor modulator	Somnolence, gastrointestinal disturbances such as diarrhea and abdominal pain, decreased appetite, and fatigue are commonly reported during cannabidiol therapy.

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
