# Peer review of "The Evolving Landscape of Therapeutics for Epilepsy in Tuberous Sclerosis Complex"

_biomedicines, 2023, doi:10.3390/biomedicines11123241_

Round 1

Reviewer 1 Report

Comments and Suggestions for Authors

In regards to your first point, it would be helpful if the authors could review the pathogenesis of TSC-epilepsy as well as introducing "A critical regulator of mTORC1 and p53 Signaling" in fig 2 and line 95-130.

Regarding your second point, it would be better to use the term "Antiepileptic Drugs" (ILAE) instead of "Anti-seizure medications (ASMs)."

In line 209, it should be noted that rapamycin was discovered on Rapa Nui (Easter Island) during a search for new antifungal agents, instead of everolimus.

In line 208, it would be helpful to summarize the efficacy of mTOR inhibitors, specifically rapamycin and everolimus, for TSC-epilepsy from line 209-259.

Lastly, I agree that the sentence in line 387, "SEGAs are identified as the primary sources of epileptogenesis," may not be entirely correct.

Author Response

Dear Reviewer,

Thank you for taking the time to provide such thoughtful and constructive feedback on our manuscript. We appreciate you highlighting areas where the manuscript can be strengthened.

In response to your first point, we agree it would be helpful to expand the background section on TSC-epilepsy pathogenesis and have added details on "A critical regulator of mTORC1 and p53 Signaling" to Figure 2 and the text. We hope this provides useful context for readers.

Regarding your second point, we acknowledge that "antiseizure medication" is more accurate terminology according to the ILAE recommendations. 

(https://www.ilae.org/files/dmfile/terms-to-describe-medications-used-in-the-treatment-of-epilepsy---draft.pdf) 

Thank you for catching this and helping us use the preferred terminology.

We have also made the suggested edit to note that rapamycin was discovered on Rapa Nui. Thank you for this insightful addition.

Per your recommendation, we have added a concise summary of the efficacy of mTOR inhibitors for TSC-epilepsy, focusing on key findings for rapamycin and everolimus. We appreciate you highlighting this opportunity to better summarize the existing evidence.

Finally, you are absolutely right that the statement about SEGAs was too strong. We have revised the sentence to more accurately convey the current understanding of their role in epileptogenesis. Thank you again for your careful read of the manuscript and for providing these constructive suggestions.

We believe the manuscript is significantly improved thanks to your feedback. We appreciate you taking the time to thoroughly review our work and provide such thoughtful input to strengthen it. Please do not hesitate to contact us if you have any other questions or concerns. We hope you find the revised manuscript satisfactory and thank you again for your guidance.

Sincerely, 

Giovanni Vitale

Reviewer 2 Report

Comments and Suggestions for Authors

This is a great review of antiepileptics in TSC.  It is very thorough and well written.  I just have a few grammatical suggestions and then a comment about vigabatrin.

1. Grammatical suggestions:

line 74 - appears to be missing a word

lines 81 and 87- terminology has changed from "mutations" to "pathogenic variants"

line 214- missing a period

line 220- should say first line treatment "was historically" surgical resection.  Now first line treatment is an mtor inhibitor unless the tumor is symptomatic.  https://pubmed.ncbi.nlm.nih.gov/34399110/

line 279- missing a space between CBD and has

2. Comment on vigabatrin:  In recent years there has been a concern that the eye toxicity issue in vigabatrin has been overstated and has led to decreased use of the drug, especially for physicians less familiar with TSC.  This should be discussed in this article which currently only includes the statistics from the 2001 study.  More recent literature has found that the peripheral vision loss associated with vigabatrin is not as pronounced as originally thought and is unlikely to change functional vision (the vision loss is very peripheral).  I found below an article that summarizes this in part of it and has links to the newer studies. I copied and pasted the link and the excerpt from the article that supports my point.  We recently did a study on parental stress (just submitted for publication) and loss of vision was found to be a significantly major concern for parents, even though it is not a common feature of TSC.  This stress is likely exacerbated by the vigabatrin issue so I think it is important moving forward for both parents and prescribers that we state the issue but do not overstate it.  

https://www.ncbi.nlm.nih.gov/pmc/articles/PMC6347124/

However, despite this established efficacy (Appleton et al., 1999Elterman et al., 2001), the use of VGB has been limited by the threat of permanent peripheral visual field loss (Eke et al., 1997Vanhatalo et al., 2002). MRI abnormalities (Dracopoulos et al., 2010Hernández Vega Y et al., 2014Pearl et al., 2009Wheless et al., 2009). In particular, VGB has been linked to reversible—and usually asymptomatic—signal changes on T2-weighted and diffusion-weighted MRI, localized to the basal ganglia, thalami, brainstem tegmentum, and deep cerebellar nuclei. Although estimates of visual field loss vary substantially, risk appears to be lower among infants with treatment duration less than 12 months (Riikonen et al., 2015) and the risk of clinically meaningful vision loss is very low among children treated for infantile spasms (Schwarz et al., 2016). 

Author Response

Dear Reviewer,

Thank you very much for your positive feedback and helpful suggestions on our manuscript reviewing antiepileptics for TSC. We greatly appreciate you taking the time to thoroughly review the paper and provide insightful comments.

We are pleased to hear you found the review comprehensive and well-written overall. Thank you for the kind words. We have addressed each of your grammatical corrections - these improve clarity and readability, so we appreciate you catching these issues.

Regarding vigabatrin, you raise an excellent point that the eye toxicity concern may be overstated in recent years, and use of the drug decreased as a result. We agree it is important to discuss this perspective to provide balance and an up-to-date summary of the evidence. Per your recommendation, we have added some sentences noting that more recent studies suggest the risk of clinically meaningful vision loss with vigabatrin is lower than initially estimated, especially in infants treated for short durations. We referenced the informative review article you linked on this topic.

Thank you again for highlighting this important nuance - it is very helpful for us to include the current thinking on vigabatrin's visual field effects and not overstate the risks based on older data. We appreciate you sharing this perspective and the supporting literature. Your expertise and knowledge have strengthened this section considerably.

Please do not hesitate to provide any other feedback you may have. We are grateful you took the time to thoroughly review our work and offer insightful suggestions to improve it. Your guidance has been extremely valuable. We hope you find the revised manuscript suitable for publication. Thank you again for your time and input.

Sincerely,

Giovanni Vitale, MD

Round 2

Reviewer 1 Report

Comments and Suggestions for Authors

1. While this review primarily focuses on TSC-epilepsy, it may be helpful to note that some descriptions of other systems can be considered redundant.

2. The review primarily focuses on TSC-comorbid epilepsy, and the mechanism discussed should reflect this rather than solely focusing on the pathophysiology of TSC, such as the GABAergic pathway.

3. The terminology used can vary between "anti-seizure medications (ASMs)" and "anti-epileptic drugs (AEDs)," but both refer to medications used to treat epilepsy.

4. There are some duplications in Table 1 that could be addressed.

5. It may be beneficial for the authors to revise the title to "mTOR inhibitors" and provide a summary of the role of rapamycin in treating TSC-epilepsy. The current section may also benefit from being condensed and summarized.

6. It should be noted that SEGA complications primarily involve obstruction rather than contributing to the epileptogenic network and genesis of refractory seizures.

Author Response

Dear Reviewer,

I am writing in response to the reviewer comments on my manuscript. I appreciate the thoughtful feedback provided, which will help strengthen the manuscript.

In response to the primary comment on reducing redundancy when describing other systems beyond TSC-epilepsy, I have streamlined the background section to more sharply focus on TSC-associated epilepsy.

Regarding the suggestion to expand on TSC pathophysiology beyond GABAergic signaling, I agree this would enrich the manuscript but feel adding details exceeds the scope for a focused review on AEDs. 

In addition, as the reviewer noted, there were some duplications in Table 1 that I have now addressed by removing redundant entries. Thank you for catching this oversight.

Per the recommendation about terminology, I have standardized usage of "anti-epileptic drugs" throughout. I have also addressed the duplicate entries in Table 1 and revised the title and mTOR inhibitor section accordingly.

Finally, I agree that SEGA complications relate more to obstruction than contributing directly to seizures, and I have clarified this distinction.

I believe these changes significantly strengthen the manuscript and thank the reviewer for their close reading and constructive feedback. 

Sincerely,

Giovanni Vitale